# The Effects of Moxifloxacin and Gemifloxacin on the ECG Morphology in Healthy Volunteers: A Phase 1 Randomized Clinical Trial

**DOI:** 10.3390/diagnostics13071234

**Published:** 2023-03-24

**Authors:** Abid Ullah, Shujaat Ahmad, Niaz Ali, Haya Hussain, Mamdouh Allahyani, Mazen Almehmadi, Ahad Amer Alsaiari, Osama Abdulaziz, Feras Almarshad, Syeda Hajira Bukhari

**Affiliations:** 1Department of Pharmacy, Shaheed Benazir Bhutto University Sheringal, Dir Upper 18000, Khyber Pakhtunkhwa, Pakistan; 2Department of Pharmacology, College of Medicine, Shaqra University, Shaqra 11961, Saudi Arabia; 3Department of Pharmacology, Institute of Basic Medical Sciences, Khyber Medical University, Peshawar 25100, Khyber Pakhtunkhwa, Pakistan; 4Department of Clinical Laboratory, Sciences Saudi Arabia Department, College of Applied Medical Sciences, Taif University, Taif 21944, Saudi Arabia; 5Department of Medicine, College of Medicine, Shaqra University, Shaqra 11961, Saudi Arabia

**Keywords:** moxifloxacin, gemifloxacin, QT prolongation, QRS-widening effect, healthy volunteers, Clinical Trial Phase 1

## Abstract

Moxifloxacin and gemifloxacin are the two newer broad-spectrum 8-methoxy-quinolone derivatives that are used to treat various bacterial infections in cardiac patients. In this research study, we assessed the impact of moxifloxacin and gemifloxacin on the QT intervals of electrocardiograms in normal adult doses and draw a comparison, in a controlled environment, on healthy volunteers. Additionally, the effect of both test drugs on the QRS complex was checked. Sixty healthy volunteers were randomly assigned to two groups via R-software, and each respectively received moxifloxacin and gemifloxacin for five days. The research ethics committee approved the research, and it was registered for clinical trial under NCT 04692623. The participants’ electrocardiograms were obtained before the start of the dose (baseline) and on the fifth day. Significant prolongation of QT interval was noted in moxifloxacin (*p* < 0.0001) as compared to gemifloxacin treated groups. There were no cases of QTc prolongation over the usual limits (450–470 ms) in the gemifloxacin-treated group, however, QTc prolongations at the rate of 30 and 60 ms from the baseline were noted, interpreted as per the EMEA guidelines. These findings indicate that moxifloxacin caused significant (*p* < 0.0001) QT interval prolongation (QTIP) as compared to gemifloxacin. In contrast to the previously reported literature, the prominent effect of moxifloxacin on the widening of the QRS-complex was noted with no such effect on QRS-widening in the gemifloxacin-treated group. It is concluded that both drugs have the potential for considerable QT interval prolongation (QTIP) effects, which is one of the risk factors for developing torsade de pointes (TdPs) in cardiac patients. Thus, clinicians should exercise caution when prescribing moxifloxacin and gemifloxacin to cardiac patients and should consider alternate treatment options.

## 1. Introduction

Fluoroquinolones (FQs) are a class of broad-spectrum antibiotics that are used to treat various infectious diseases [1]. The discovery of nalidixic acid brought about the era of quinolones [2]. Over the past two decades, quinolones have evolved from a relatively small and unimportant group of antibiotics to a large, important, and predominant group of antibiotics used to treat respiratory and urinary tract infections [3]. Currently, the newer fluoroquinolones have an ideal market position in comparison to previous antibiotics, owing to their wide antimicrobial spectrum coverage, enhanced effectiveness, better pharmacokinetics, single daily dosage and few drug–drug interactions [4,5]. Apart from their wonderful commercial position, the FQs were a major subject of discussion among pharmacists and doctors, owing to certain hitherto unknown adverse effects [6]. The prolonged use of FQs in elderly cardiac patients has prompted serious concerns regarding their medical safety profile [7,8,9]. According to several case study reports and pharmacovigilance data, fluoroquinolones have been associated with blood glucose abnormalities [10,11]. Fluoroquinolones have also been linked to cardiovascular toxicity, including QT prolongation and torsade de points (TdPs) [12]. This QT represents both depolarization and repolarization of the heart ventricle action potential [13]. Because the QT interval is often shortened in tachyarrhythmia and prolonged in bradyarrhythmia, the QTc interval is used for actual calculation [14]. TdPs is a risk factor for cardiac toxicity and is one of the most frequently seen clinical concerns that are resultantly reported with ciprofloxacin clinical use [15]. The TdPs is a ventricular repolarization anomaly that is preliminarily characterized by the twisting of points on ECGs [16]. Fluoroquinolones carry a higher risk of causing TdPs [17], which is why several FQs have been withdrawn from the market [3,18].

Considering the widespread use of fluoroquinolone antibiotics in cardiac patients, we evaluated moxifloxacin and gemifloxacin in normal adult doses for probable QT interval prolongation via in vivo techniques on healthy volunteers in a controlled environment.

## 2. Materials and Methods

### 2.1. Study Design

The Pharmacy Department followed the good clinical practice principles in recruiting healthy volunteers for the clinical trial. One group was given moxifloxacin, while the other received gemifloxacin. On days 1 and 5, ECGs were obtained to determine the QT interval. Food, drinks and fluids were maintained at constant levels throughout the experiment. The volunteers were properly informed by the principal investigator regarding the project aim, procedures and risks, and written consent was given prior to participating in the trial. The study was approved by the Shaheed Benazir Bhutto University’s Ethical Committee (Approval no SBBU/IEC-20-01) and registered with Clinical Trials.gov as NCT 04692623.

### 2.2. Participants

Prior to enrollment, all participants were informed of the study’s risks and benefits. Their vital indicators, such as blood pressure and electrocardiograms, were assessed accurately and informed consent forms were completed by all participants.

#### 2.2.1. Inclusion Criteria

Included participants were shown to be healthy and non-smokers between the ages of 20–40 years and free from any type of cardiac disease.

#### 2.2.2. Exclusion Criteria

Participants were excluded:Whose ages were age <20 years or >40 years, due to increased risk with advancing age. Similarly, old females at twice the risk were excluded.Hypocalcemia, hypokalemic and participants suffering from ischemia and LV dysfunction.Participants having QTc ≥ 450 ms before starting the dose, QRS ≥ 120 ms, with a family history of cardiovascular disease (CVDs), heart beats <50 BPM or >100 BPM, or with a history of fluoroquinolones allergy were also excluded.Female participants who were lactating or pregnant were not selected.

### 2.3. Safety Monitoring and Trial Discontinuation

The clinical trial protocols were followed as per EMEA guidelines where the safety of the participants was assured, and participants were discontinued from the trial if they faced any undesirable adverse effects.

### 2.4. Sample Estimation and Randomization

Per the standing practices of FDA guidelines, 10–50 volunteers may be enrolled in a phase 1 clinical trial. However, on a 95% Confidence Interval, with a marginal error of 1 and 50% variability, the sample size was estimated as 50. To this 50, we added 20% (10) attrition. Thus, a total sample of 60 volunteers was enrolled [19]. The participants were divided into two equal groups through statistical R software version 7.3. Each group consisted of 30 participants. A total of 50 participants completed the trial and 10 discontinued the trial; the data of these 50 participants were analyzed. These were allotted specific code numbers to avoid biases at the time of distribution into two groups. The sample size was based on the power calculation, based upon previous studies suggesting this for randomized clinical trials and research studies utilizing sample sizes that are too small to ensure statistical significance results. Therefore, in the current study, we could register n = 60 subjects to reach our conclusion.

### 2.5. Procedures

#### 2.5.1. Drugs and Dosing

The drugs (moxifloxacin (400 mg) and gemifloxacin (325 mg) used in this study were of standard pharmaceutical firms and were given orally to the participants of both groups in once-daily dosing for five days.

#### 2.5.2. Instruments

Sphygmomanometer was used for the measurement of blood pressure on day 5, following an oral administration of the drugs. For ECGs recording (12-Leads) ECG machine was used.

##### Blood Pressure Measurements

Blood pressures (BP) of the subjects were determined by sphygmomanometer. BP (systolic and diastolic) was checked in terms of mm Hg in a sitting position, after taking rest for 10 minutes in supine position.

##### ECG Recording

Since our target was not to determine either concealed prolonged QT interval or those QT interval prolongations which are evident in certain favorable conditions, we did not use Holter Monitors [20]. Our aim was to determine the direct effects of the test drugs, therefore, we used standard ECG machine to record the QT interval prolongation. A standard 12-leads ECG machine was used for the participants’ ECGs recording at the physiology laboratory (BMS-Lab) of the Pharmacy Department. Before the ECGs recording, all the participants were directed to rest quietly for 10–15 min in the supine position. Before the administration of the drug by the volunteers, the baseline ECGs (Pre-dose ECG) were obtained in triplicate and then a second set of ECGs were obtained after the administration of the drug on the fifth day, with triplicate recording after a 1 min interval.

(a)ECG Analysis and Interpretation

Since the QT interval is often shorted in tachyarrhythmia and prolonged in bradyarrhythmia, the QTc interval is used for actual calculation, and for that basis we used Bazett’s formula to calculate the QTc.

Manual analysis and interpretation of all ECGs were determined, and their respective corrected QTc were measured [21].

Since different formulae are used to analyze the QTc, and as our target population did not include any adolescents, therefore, we used the Bazett’s Formula as reported by Goldenberg et al., 2006 [22].
(1)$$\mathrm{QTc}=\frac{\mathrm{QT}}{\sqrt{\mathrm{RR}}}$$

The borderline QTc range is from 431 to 450 ms in males and 451 to 470 ms is the borderline QTc range in females [23]. QTc intervals ranging from ≥450 ms in males and ≥470 ms in females were considered abnormal [24].

The QRS complex was also calculated while taking the QRS complex normal value ranges from 60 to 110 ms [25].

(b)QRS-Complex Measurements

The QRS complex is measured as the end of the PR-interval or the beginning of the Q wave to the end of S-wave [25].

(c)Blinding

The analysis and interpretation of all ECGs were performed by a well-experienced cardiologist who was blind of study objectives and ECGs replicate numbers. The cardiologist then verified the interval durations and performed the morphological analysis of all ECGs.

### 2.6. Adverse Drug Effects Monitoring

During the trial, adverse effects were noted on the proforma designed for the purpose (attached as Appendix A).

### 2.7. Statistical Analysis

Graph Pad Prism was used for the statistical analysis of collected data. The difference in QT intervals on the baseline and on day five in both drug-treated groups was extracted by one-way analysis of variance (ANOVA) test and paired sample *t*-test. *p*-values < 0.0001, < 0.001 and < 0.05 were expressed as ****, *** and **, respectively, and were considered statistically significant.

## 3. Results

### 3.1. Effects on QT Interval

According to the participant demographics, a total of sixty-five participants were screened, out of which five failed to meet the eligibility criteria. The sixty volunteers were equally randomized into two groups comprising thirty participants respectively. Of the sixty participants, fifty were male and ten were female. Twenty-five males and five females were allotted numbers, after randomization, to moxifloxacin and gemifloxacin groups, respectively.

The mean ages of the individuals in both groups were 27 and 31 years, respectively. The mean baseline blood pressures measured in both groups (moxifloxacin and gemifloxacin) were 120.8/79.6 and 116.8/78.8 mm Hg, respectively.

Table 1 shows the demographic characteristics of participants, and Figure 1 depicts the study’s follow-up of volunteers. In both groups, ten participants terminated the study due to severe adverse drug reactions (skin rashes), COVID-19 emergency, and illness progression. The subjects’ ECGs were analyzed to determine QTc Prolongation. The data indicates that four out of twenty-five treated volunteers (*n* = 4) had a prolonged QTc (450 ms for males) in the moxifloxacin group, with a baseline QTc of 415.9 ± 31.5 ms and a mean QTc of 455.9 ± 11.8 ms on the fifth day. As shown in Figure 2, the mean QTc of total twenty participants in the moxifloxacin-treated group was 385.88 ± 29.21 ms on the baseline (day 1st). On the fifth day, the QTc was 413.32 ± 31.49 ms. On the first day, the baseline QTc was 351.0 ± 48.2 ms, and on the fifth day, it was 379 ± 41.9 ms in the gemifloxacin-treated group, as shown in Figure 2.

There was one case in the moxifloxacin group (*n* = 1) where the QTc was within the normal range but was 60 ms longer than the baseline value, 348 ms on the baseline and 413 ms on the fifth day, and two cases in the gemifloxacin group (*n* = 2) where the QTc values were 300 ms and 347 ms on the baseline and 359 ms and 447 ms on the fifth day, respectively.

Results for the effects of the test drugs on each gender, and the implications in perspective of EMEA guidelines, are plotted in Table 2. The moxifloxacin-treated group in both males and females showed QTc prolongation from the baseline normal ranges, according to the EMEA guideline ranges. In the gemifloxacin group, males observed significant QT prolongation of more than 60 ms from the baseline on day five.

There was no evidence of QTc prolongation above the prescribed limits (450–470 ms) for males or females in the gemifloxacin-treated group (*n* = 25) as presented in Table 2. Overall data of QTc of twenty participants separately of both moxifloxacin and gemifloxacin treated groups are presented in Table 3 and Table 4 respectively, while the mean QTc of all 25 participants both treated groups in tabulated in Table 5. Additionally, the effect of both test drugs on the QRS complex was checked. Five males and one female participant in the moxifloxacin-treated group had a wider effect on the QRS complex as of day five when compared to their baseline QRS complexes, as shown in Figure 3. Moxifloxacin induced a widening of the QRS-complex in the moxifloxacin-treated group, but gemifloxacin had no significant impact. The ST segment and flattening of the T wave were not noted in our study tests, but a broadening of the QRS complex was noted, which contributed to the QT prolongation effect. This suggests that moxifloxacin slows down the depolarization phase of the action potential of the heart.

### 3.2. Adverse Effects

Throughout the clinical study, many moderate adverse effects were observed, including four patients experiencing hypotension, eleven experiencing dizziness and lightheadedness and four experiencing nausea and vomiting in the moxifloxacin group. Similarly, four participants in the gemifloxacin group faced skin rashes, as shown in Table 6.

## 4. Discussions

Moxifloxacin and gemifloxacin are currently used to treat a variety of illnesses, owing to their therapeutic significance and better clinical profile. They are often discontinued due to adverse consequences. The most common of these adverse effects is QT prolongation [26,27]. As a result of cardiotoxicity and QT prolongation, grefafloxacin and sparfloxacin were withdrawn from the market [8,28]. In this study, one participant in group-A (moxifloxacin) and two participants in group-B (gemifloxacin group) reported the prolongation of QTc from baseline at the rate of >60 ms. These findings are consistent with earlier reports that QTc prolongation of 6–16 ms with moxifloxacin carries a substantial risk of developing TdPs [17,29]. Similarly, four cases (three males and one female) in the moxifloxacin-administered group had respective QTc values of 436,390,388 and 449.43 ms on day one, and 450,449.53, 450.57 and 473.68 ms on day five. Other studies suggest that moxifloxacin caused reversible blockade of the K_ir_ potassium channels present in the heart and thereby caused a delay in the internal rectifying current. After 2–4 h of moxifloxacin administration, this resulted in a QT interval prolongation of more than 60 ms from the baseline [30]. Thus, our findings are consistent with other reported work. The more frequent occurrence of prolongation with moxifloxacin in healthy volunteers raises concern about its safety, particularly in the elderly, the most vulnerable patients.

On day five, one participant in the moxifloxacin and two participants in the gemifloxacin group had a QTc prolongation of more than 60 ms, indicating an increased risk of torsade de pointes (TdP) [31]. These four cases of QTc prolongation in the moxifloxacin group revealed in our investigation substantially corroborate the previously published cohort study by Menon V. et al., 2010 [32]. According to these investigations, fluoroquinolones were the most frequently given medication that resulted in hospitalization for QT interval prolongation [33].

The maximum moxifloxacin and gemifloxacin concentrations in blood and the time of ECGs replication are critical because they have a direct influence on QTc prolongation [34], which is why we recorded the second replicate of ECGs on day five as Abels et al., 2001 described. In our investigation, moxifloxacin had a much greater impact on QTc interval prolongation than gemifloxacin. This might be because moxifloxacin has a larger impact on prolonging ventricular repolarization than gemifloxacin, which is why caution signals have been added to the moxifloxacin leaflet [35]. Moxifloxacin’s impact on QT prolongation was, for the first time, attributable to a broadening of the QRS complex, with no effect on the ST-segment and flattening of T-wave, as shown in Figure 3. Our findings substantially align with prior research, since we observed a greater impact of moxifloxacin on *QTc* than gemifloxacin in our study [36]. The QRS complex-widening effect caused by moxifloxacin is the ultimate cause of QT prolongation, as reported in this study. This effect is due to the sodium influx blockade via sodium–potassium ATPase pump, and thus causes a widening effect on the QRS [37]. As far as the selected fluoroquinolones (Moxifloxacin and gemifloxacin) are concerned, they are targeting the repolarization phase and caused QT interval prolongation in the ECG morphology and have no effects on the depolarization phase [38,39].

In this study, the results show that only moxifloxacin showed a QRS-widening effect, with none in the gemifloxacin-treated group, as shown in Figure 3, Table 7 and Table 8.

This QT interval prolongation effect of moxifloxacin may cause a blockade of K_IR_ heart channels via binding with the S6 pore domain of human ether-a-go-go-gene related protein (hERG) potassium channels of the heart by binding to its Tyr652 residue site [39].

This raises severe concerns about the safety of moxifloxacin, particularly for cardiac patients who are currently taking medications for cardiovascular problems and other clinical ailments that require antibiotic therapy. Additionally, an in vitro investigation showed that several fluoroquinolones had a substantial impact on HERG channels [39]. Noel G. et al., 2001 performed research to determine the comparative impact of various FQs on QTc prolongation (Ciprofloxacin, levofloxacin and moxifloxacin). The finding indicates that moxifloxacin had a similar impact to ciprofloxacin and levofloxacin [33], indicating the need for closer attention to their possible effects on ECGs. Similarly, we did not observe significant changes in QTc with gemifloxacin and, therefore, our results are consistent with previous studies that report no significant changes in QTc with gemifloxacin [40]. We performed experiments to determine the effects of gemifloxacin on QT interval prolongation effects, interpreted as per the EMEA guidelines, with more prominent effects from moxifloxacin in both genders. However, gemifloxacin observed a positive QTc prolongation effect in males. We could not determine the effect on the females’ ECG, as we had no female subjects. However, overall, no substantial QT prolongation was seen in our research investigations with gemifloxacin as compared to moxifloxacin. The same weak effect was reported for gemifloxacin as compared to moxifloxacin in our previous study on potassium channels in pancreatic beta cells [41]. The results of our study compared well with the effects of moxifloxacin reported previously, where a retrospective analysis of pooled data from QT studies with moxifloxacin as a single 400 mg dose in healthy volunteers was studied [42].

Maximum QTc prolongation effects achieved with moxifloxacin are in line with the previous studies conducted, which concluded that moxifloxacin could induce ventricular repolarization [43]. The QTc effects of moxifloxacin and gemifloxacin observed in our study in our targeted volunteers’ population are consistent with available data designed for QT studies. The effects on QRS were more prominently targeted in our study, which was lacking in previous studies conducted for the QT-prolongation effect of moxifloxacin and gemifloxacin [44,45,46,47].

Upon translation, the results of this study indicate that, in clinical scenarios—especially in cardiac and arrhythmic patients using anti-arrhythmic drugs—prescribing moxifloxacin could lead to severe cardio-toxicity due to QT interval prolongation.

## 5. Limitations

The results of this study, particularly the adverse effects profile, require further studies in a larger population for the safe practice of medicine. We did not use a Holter monitor for 24 h follow-up, which could have been ideal for comparison. Therefore, further studies are required to determine the mechanisms of the QT prolongation effect of these drugs by using the Holter monitor. Secondly, we used Bazett’s formula, as it is from standing practice that for drugs already in the market or already in clinical practice. The Bazett’s formula is considered as the most widely used for QTc correction method, and for new drug screenings, the Fridericia formula is for new drug clinical trials to check their safety as recommended by the U.S. Food and Drug Administration (FDA). A relatively small number of female healthy volunteers (*n* = 10) due to unavailability, limited the opportunity to investigate the effects in females with adequate statistical power. Accordingly, we added this as a limitation of the study.

## 6. Conclusions

The present study concluded that moxifloxacin caused significant QT interval prolongation as compared to gemifloxacin. Thus, clinicians should exercise caution when prescribing moxifloxacin and gemifloxacin to cardiac patients and should consider alternative treatment options.

## 7. Recommendations

Clinicians shall remain vigilant while prescribing moxifloxacin and gemifloxacin to cardiac patients where moxifloxacin and gemifloxacin are required for the treatment of other concurrent chief complaints. It is, therefore, recommended to roll out another substitute antibiotic based on the clinical rationale. It is better to use other antibiotics for people with heart disease.

## Figures and Tables

**Figure 1 diagnostics-13-01234-f001:**
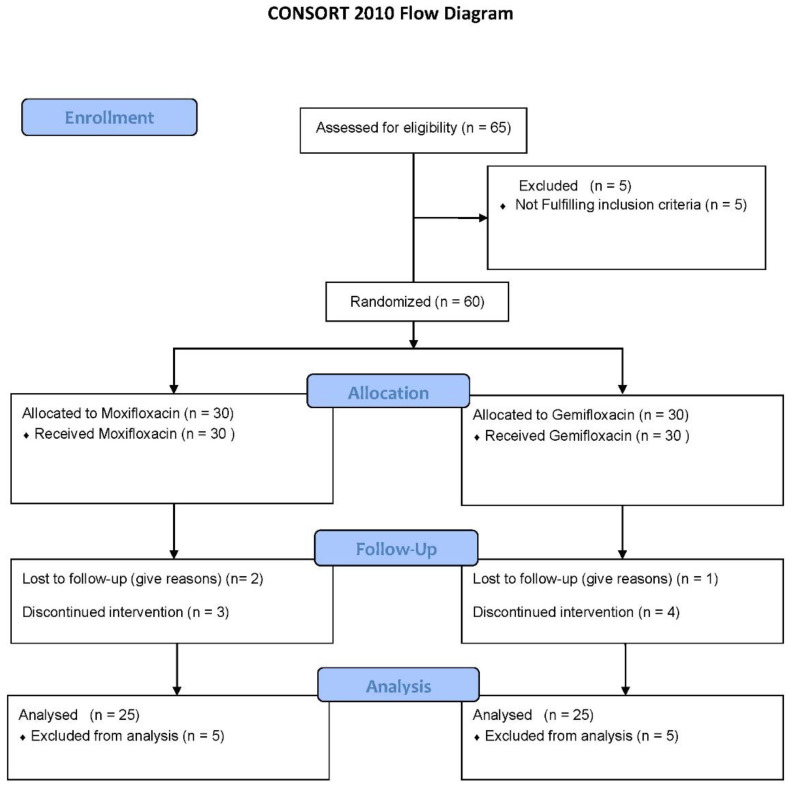
Clinical Trial Profile.

**Figure 2 diagnostics-13-01234-f002:**
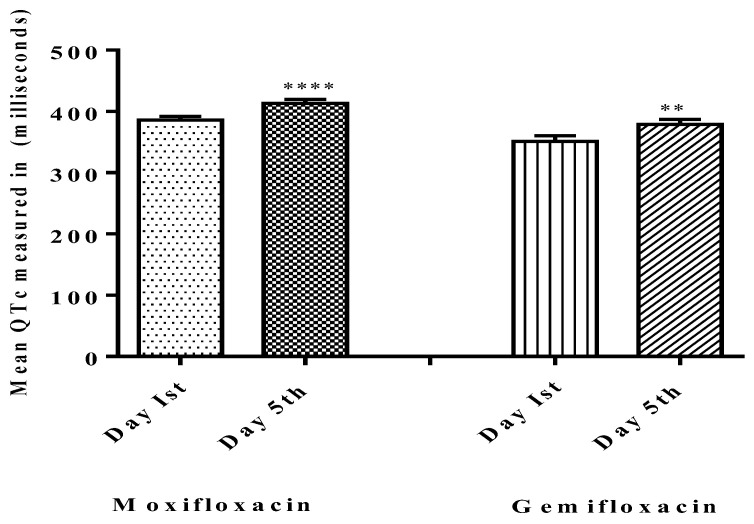
Comparisons of QTc on baseline day one (Pre-dosing) and after dosing day five (Post-dosing) in both groups. Significant differences in QTc in the moxifloxacin-treated group were noted and gemifloxacin was compared on day one and day five. **** shows that *p* < 0.0001 which shows high level of significance by Moxifloxacin as compared to Gemifloxacin ** *p* < 0.05.

**Figure 3 diagnostics-13-01234-f003:**
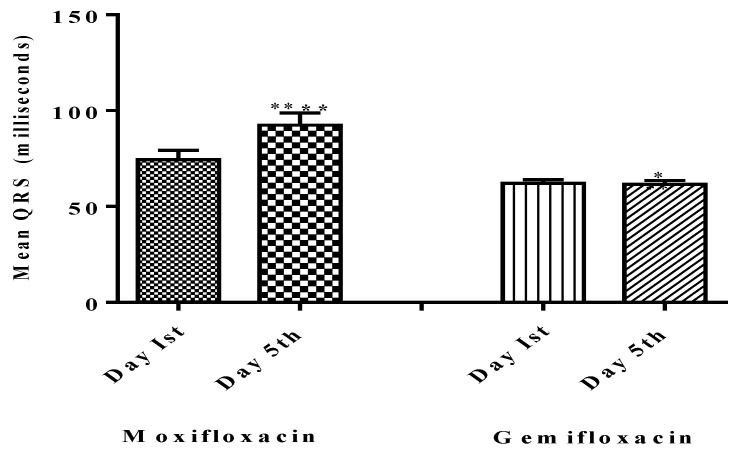
Comparison of QRS on baseline day one (Pre-dosing) and after dosing day five (Post-dosing) in both groups. Significant differences in QRS Complex in the moxifloxacin-treated group were noted, as compared to gemifloxacin. **** shows that *p* < 0.0001 which shows high level of significance by Moxifloxacin as compared to Gemifloxacin * *p* < 0.05.

**Table 1 diagnostics-13-01234-t001:** Baseline characteristics of the participants plotted as mean ± SD.

Demographic Data	Moxifloxacin	Gemifloxacin
Age in Years	27.4 ± 2.9	31.4 ± 5.4
Male/Female Ratio	25/5	25/5
Height in Cm	164 ± 5.1	165.3 ± 5.8
Weight in Kg	68.16 ± 5.1	67.4 ± 4.1
Basal Mass Index (BMI) (Kg/m^2^)	23.9 ± 1.7	24.0 ± 1.9
Body Temperature (°C)	37.1 ± 0.3	37 ± 0.3
BP Systolic/Diastolic (mm Hg)	120.8 ± 10/79.6 ± 4.5	116.8 ± 7/78.8 ± 3.3
Pulse (BPM)	77.4 ± 5.7	78.4 ± 6.7

**Table 2 diagnostics-13-01234-t002:** Gender effect of moxifloxacin and gemifloxacin on QTc interval.

Drug Group	QTc Mean (ms) in Male Treated Group (Mean ± SD)		Mean QTc (ms) in Female Treated Group (Mean ± SD)	
	Pre Dosing Day 1st	Post Dosing Day 5th	Pre Dosing Day 1st	Post Dosing Day 5th
Moxifloxacin	404.7 ± 7.1 (*n =* 3)	450 ± 0.57 (*n =* 3)	449.4 (*n =* 1)	473.7 (*n =* 1)
Gemifloxacin	323.5 ± 33.2 (*n =* 2)	393.5 ± 48.9 (*n =* 2)	(*n =* 0)	(*n =* 0)

**Table 3 diagnostics-13-01234-t003:** QTc interval measured in milliseconds in twenty-five participants treated with Moxifloxacin.

Fluoroquinolone Used	Baseline ECG on Day 1	ECG on Day 5
	QTc in ms	QTc in ms
Moxifloxacin (*n =* 25)	351	384
	415	418
	421	433
	366	409
	372	413
	380	415
	371	373
	378	382
	416	444
	370	410
	377	400
	388	450.57
	436	450
	390	449.43
	416	444.4
	398	418
	401	400
	380	424
	370	410
	351	384
	348	413
	359	386
	393	418
	415	444
	449	473.68

**Table 4 diagnostics-13-01234-t004:** QTc interval measured in milliseconds in twenty-five participants treated with Gemifloxacin.

Fluoroquinolone Used	Baseline ECG on Day 1	ECG on Day 5
	QTc in ms	QTc in ms
Gemifloxacin (*n =* 25)	385	400
	386	388
	272	300
	324	385
	416	423
	300	359
	387	400
	300	342
	336	359
	363	368
	360	381
	433	463
	347	428
	405	404
	342	379
	300	340
	380	413
	282	320
	424	424
	346	388
	369	375
	360	377
	351	379
	386	388
	324	385

RR and QT were calculated and were converted into QTc by the Bazett’s formula as mentioned in the study design.

**Table 5 diagnostics-13-01234-t005:** Baseline QTc (ms) and QTc (ms) on day 5th in Moxifloxacin and Gemifloxacin groups.

Fluoroquinolones	Mean QTc on Day 1	Mean QTc on Day 5	*p* Value
Moxifloxacin (*n* = 25)	385.8 ± 29.2	413.3 ± 31.4	*p* < 0.0001 **
Gemifloxacin (*n* = 25)	351 ± 48.2	379 ± 41.9	*p* < 0.05

** Shows a significant effect on QTc changes by Moxifloxacin as compared to Gemifloxacin. Data extracted via (one-way ANOVA).

**Table 6 diagnostics-13-01234-t006:** Adverse Drugs reactions noted during the clinical trial.

Event/ADEs	Moxifloxacin (*n* = 25)		Gemifloxacin (*n* = 25)	
	Frequency	Percentage	Frequency	Percentage
Hypotension	4/25	16	0/25	0
Nausea and Vomiting	4/25	16	0/25	0
Skin flushing	0/25	0	04/25	16
Lightheadedness and Dizziness	11/25	44	0/25	0
Diarrhea	4/25	16	0/25	0

**Table 7 diagnostics-13-01234-t007:** Baseline QRS (ms) complex and QRS (ms) complex on day five in the Moxifloxacin group.

Fluoroquinolone Used	Baseline ECG on Day 1st	ECG on Day 5th
	QRS in ms	QRS in ms
Moxifloxacin (*n =* 25)	60	60
	60	60
	40	40
	80	80
	60	120
	60	60
	80	100
	80	100
	100	120
	120	140
	120	120
	100	120
	80	120
	80	140
	100	100
	100	100
	60	60
	60	60
	80	120
	80	120
	80	90
	60	60
	60	60
	80	120
	80	80

**Table 8 diagnostics-13-01234-t008:** Baseline QRS (ms) complex and QRS (ms) complex on day five in the Gemifloxacin group.

Fluoroquinolone Used	Baseline ECG on Day 1st	ECG on Day 5th
	QRS in ms	QRS in ms
Gemifloxacin (*n =* 25)	60	60
	60	80
	60	60
	60	60
	80	80
	60	60
	60	80
	60	40
	80	60
	60	60
	60	80
	60	80
	60	60
	60	80
	80	60
	60	80
	80	60
	60	60
	60	60
	60	60
	60	60
	60	60
	60	60
	60	80
	60	60

## Data Availability

The datasets (ECGs of participants) for the current study are available from the corresponding authors on reasonable request.

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
