# Peer review of "The Effects of Moxifloxacin and Gemifloxacin on the ECG Morphology in Healthy Volunteers: A Phase 1 Randomized Clinical Trial"

_diagnostics, 2023, doi:10.3390/diagnostics13071234_

Round 1
Reviewer 1 Report
Thanks for giving me the chance to review such an interesting study. Really, the manuscript is well written, but, the folllowings should be considered:
The authors mentioned that an questionnaire was conducted during the study related to the side effects, is this questionnaire validated and the questions were not written in the manuscript.so, the authors should clarify this questionnaire well
Author Response
Thanks for yours worthy comments.Response to yours worthy comments are herby attached.

Reviewer 2 Report
The study is conducted using Healthy Volunteers to check the cardiovascular systemic (CVS) affects of Moxifloxacin and Gemifloxacin. Both drugs are in use for a long time and their CVS effects are well known. In the present, the researchers investigated the CVS effect which is known. In the study design, it was given actual Enrollment is 25 participants, but its appearing 60 subjects in a study. The gender of the subjects includes females. In exclusion criteria, it was stated that old females at twice the risk were excluded. How the female healthy individual will be suitable for this study. The hormone levels may play a role in the CVS effect. The novelty of the study should be highlighted. How the current study is deferring from the existing available reports. The flow chart should be corrected. Also, the summary of the study findings can be updated in 'https://clinicaltrials.gov/ct2/show/study/NCT04692623?cond=NCT+04692623&draw=2&rank=1'.
Author Response

(The authors gave the same response as above.)

Reviewer 3 Report
Key: question to authors:
The data and ECG parameter (HR, QRS-duration, QT-interval and QTCB interval) are not rightly represented in the current version. Need an extra tables to show all ECG parameters.
In the current table 3 (key):
1). number of patients are missing;
2). data are presented in MEAN+/- SD?
3). if QTcb-interval is expressed by % changes after treatment, then
moxifloxacin prolonged QTcB by 11% in male and only by 5% in female group
while gemifloxacin prolonged QTcB by 21.5% from its baseline value in male group (data obtained from Table 3).
The authors need to take the difference baseline value of ECG parameters (QT, QTcB and QRS duration) for data analysis.
These data indicate that gemifloxacin prolonged QTcB more than that in moxfloxacin, which does not support the conclusion of the manuscript.
The data also did not support that female gender are more susceptible than men to drug-induced QT prolongation (Borje Darpo et al., 2014).
Author Response

(The authors gave the same response as above.)

Round 2
Reviewer 2 Report
The manuscript is modified as suggested and can be accepted for publication.
Author Response
Thank you worthy reviewer for yours comments.We revised the manuscript accordingly.

Reviewer 3 Report
The data and conclusion are still needed to updated.
1). the baseline value of QTc-interval was higher in the Mox-treated group compared to that in Gem-treated group, and the outcome of QTc-prolongation above 450 msec was n=2 out of 25 in the Mox-treated group vs 0 out of 25 in the Gem-treated group: that is not significant nor relevant enough to put into the conclusion.
2). the statement of the slowing the conduction time (QRS-duration): seems to me there is no enough date to to support it, since the baseline value are different in these two groups.
3). statistical comparison between the two groups should be done considering the different baseline values in the two groups: delta or % changes may be more accurate than the actual Unit .
4). Details of QRS-duration data are needed to presented as well if the data support the conclusion.
5). sample size of gender difference in Table is too small to present in the manuscript n=1 or 3.
Author Response

(The authors gave the same response as above.)

Round 3
Reviewer 3 Report
ok now